# Quantifying tourism booms and the increasing footprint in the Arctic with social media data

Claire A. Runge [ID]¹*, Remi M. Daigle², Vera H. Hausner¹

**1** Arctic Sustainability Lab, Department of Arctic and Marine Biology, UiT The Arctic University of Norway, Tromsø, Norway, **2** Département de biologie, 'Université Laval, Québec, Canada

* claire.runge@uqconnect.edu.au, twitter @Claire_Runge

## Abstract

Arctic tourism has rapidly increased in the past two decades. We used social media data to examine localized tourism booms and quantify the spatial expansion of the Arctic tourism footprint. We extracted geotagged locations from over 800,000 photos on Flickr and mapped these across space and time. We critically examine the use of social media as a data source in data-poor regions, and find that while social media data is not suitable as an early warning system of tourism growth in less visited parts of the world, it can be used to map changes at large spatial scales. Our results show that the footprint of summer tourism quadrupled and winter tourism increased by over 600% between 2006 and 2016, although large areas of the Arctic remain untouched by tourism. This rapid increase in the tourism footprint raises concerns about the impacts and sustainability of tourism on Arctic ecosystems and communities. This boom is set to continue, as new parts of the Arctic are being opened to tourism by melting sea ice, new airports and continued promotion of the Arctic as a 'last chance to see' destination. Arctic societies face complex decisions about whether this ongoing growth is socially and environmentally sustainable.

## Introduction

The number of tourists visiting the Arctic has increased dramatically over the past two decades [1,2], reflecting a rise in tourism globally over the past 50 years [3]. While this could bring alternative livelihoods to Arctic communities, concerns over the social and environmental sustainability of the rate and scale of the tourism boom are growing across the Arctic [4–6]. Tourism can have both positive and negative impacts on the natural environment and on host communities. Direct effects of tourism (e.g. transporting, accommodating, and feeding tourists) and the indirect socioeconomic change brought about by the tourism industry (e.g. influx of seasonal workers) drive increased habitat loss [7], resource use, and carbon emissions [8] across the world, in addition to the localized consequences of nature-based tourism and recreation activities on the natural environment, such as injury, death, or disturbance of wildlife or damage to vegetation [9]. Understanding, at a local scale, how and where tourism booms are

the footprint across time publicly and freely available for download at doi:10.18710/QEOFPY.

**Funding:** The work was funded by grants awarded to VHH from FRAM - High North Research Centre for Climate and the Environment through the Flagship MIKON (Project RConnected; https://www.framcentre.com/) and the Arctic Belmont Forum Arctic Observing and Research for Sustainability (Project CONNECT; https://www.belmontforum.org/). The Norwegian collaboration was financed by Norwegian Research Council grant 247474 (https://www.forskningsradet.no/en). The funders had no role in study design, data collection and analysis, decision to publish, or preparation of the manuscript.

**Competing interests:** The authors have declared that no competing interests exist.

distributed across landscapes is crucial for conserving Arctic environments and for managing impacts on host communities.

A major challenge for planning and managing sustainable tourism growth in the Arctic lies in the difficulties of mapping where tourists go and how they use landscapes and ecosystems. Data on spatial visitation and trends is sparse in the Arctic. While statistics on hotel stays and transport use are now commonly collected by government and tourism management organizations, data on where tourists go during the day and what they do there is rarely collected.

Social media provides a useful source of information on tourist visitation patterns to better pinpoint the needs of tourists and target actions to manage tourism impacts [10]. Passively crowdsourced and high resolution information from social media (volunteered geographic information; VGI) can be used to rapidly generate maps of the multiple destinations visited by tourists across large areas and over time [11]. Social media data has been well demonstrated to be useful for mapping and monitoring at a range of spatial scales across the world. Such data has been used to map the distribution of tourists in protected areas [12–14], and within cities [11] and can be used to inform a variety of aspects of tourism and landscape management, including to identify peaks of visitation to attractions [15,16], map environmental impacts [17] and to estimate the landscape values, nature-based experiences and ecosystem services appreciated by tourists [18–20]. Most of these studies use the social media platform, Flickr, which has been shown to correlate well with visitor statistics at different scales [12,15,16,21].

Similar to many nature-based tourism destinations in developing countries, parts of the Arctic in Scandinavia, Iceland, Faroe Islands and Alaska have experienced unprecedented growth in the number of tourists in recent years, and Greenland, Canada and the Russian Arctic are likely to be the next tourism frontiers [22,23]. Rapid and localized booms in tourism can overwhelm local capacity (and desire) to host visitors, particularly in small, remote communities such as those found in the Arctic and many parts of the developing world [23–26]. The ability to rapidly identify sites in the early stages of a boom would support better adaptation and planning to pre-emptively address many of the sustainability challenges brought about by rapid increases in nature-based tourism. It would allow local communities and national tourism organizations to proactively direct resources to sites where special management such as provisioning of restrooms and waste disposal, better signage, safety and disaster management, parking, and trail maintenance may be required. Social media data has been demonstrated as an early-warning system for rapidly detecting booms and busts in such diverse applications as disaster management [27,28] and disease control [29]. Methods for such 'event detection' are rapidly evolving [30,31]. These methods rely on large amounts of high temporal resolution data ('big data'). The suitability and limitations of social media data for detecting events in data-sparse regions has yet to be tested. We investigate whether Flickr data can be used as an early warning system to detect localized booms in tourism in the Arctic and similar data-deficient regions.

Tourism growth can influence the spatial use of landscapes in various ways. Here, we demonstrate how the tourist footprint on Arctic landscapes has changed over the past 14 years at a pan-Arctic scale, and examine the management implications of those changes. We test two hypotheses drawn from theories of tourism and economic geography [32,33] 1) that tourists visit the same sites through time (overall footprint is unchanged but magnitude of use at each site has increased) 2) that tourists have spread throughout the landscape (overall footprint increased but magnitude of use at any one site is constant). These two patterns of tourism growth have very different implications for both the social and environmental sustainability of tourism and the satisfaction of the tourist experience. For instance, if tourists avoid busy areas and self-segregate across landscapes [34], environmental and social impacts will be more widespread but lower intensity. Alternately, the negative impacts of tourism can be localized by

channeling tourists into high use sites and away from sensitive communities and ecosystems. We examine seasonal differences in the spatial patterns of Arctic visitation, and explore how infrastructure such as roads, airports and ports influences these patterns, with a view to informing how upcoming infrastructure development could contribute to tourist growth.

## Materials and methods

### Extraction of data from Flickr

We first extracted geotagged and publicly shared photo metadata for over 2 million photos from Flickr (www.flickr.com) for the region north of latitude 60˚N. Photo metadata included location and date that each photo was taken, user id (key coded by Flickr), image URL, Flickr- and user- generated image tags, and user-generated image title. Data was extracted from the Flickr API (https://www.flickr.com/services/api/) on 4 December 2017. Due to an issue with the data download we re-extracted photos for Iceland (bounded by -27˚ to -12˚ longitude and 62˚ to 68˚ latitude) on 11 January 2018. Hourly metrics of the number of photos uploaded to Flickr globally between to January 1 2004 and December 31 2017 were obtained from the Flickr API on 07 May 2018. We used the R package 'flickRgeo' [35] which provides an R wrapper for the Flickr API.

For the purposes of this study we define our study region, 'Arctic', as the region bounded by the Arctic Council AMAP boundaries [36] and confined to areas north of latitude 60˚N. We constrain the study region using an environmental rather than political definition as the study is focused primarily on impacts on Arctic landscapes. We excluded photos from the extracted dataset that were taken outside this study region. We also excluded photos that were missing urls or geotag coordinates, had null coordinates (0,0), and photos taken prior to January 1 2004, or after December 31 2017. We excluded photos by users with only 1 or 2 photos within the study region as they are likely to represent people who are just trialing Flickr by uploading a random photo rather than a photo representing a genuine tourist. These 'test users' account for approximately 36% of users in the Flickr dataset but just 0.95% of photos. Further details on the choice of 2 photos as a threshold for exclusion are included in S1 Appendix. The final dataset contained a total of 805,684 geotagged photos with metadata from 13,596 unique users.

### Mapping the seasonal distribution of Arctic tourism

To map the relative intensity of visitation across the Arctic in summer and winter, we first created square spatial grids (rasters) at 10km and 100km resolutions. We then calculated the photo-unit-days in each grid cell for summer and for winter aggregating data for each season across all years (2004–2017). We defined the months of May to October as 'summer' and November through April (of the following year) as 'winter' (e.g. "winter 2016" includes the months November and December in 2016 and January through April in 2017). Photo-unit-days (PUD) is an established metric of tourism visitation [15] that accounts for the biases in social media data introduced by differences in the number of photographs uploaded by different users. For example, three PUD can represent either a site visited on three separate days throughout the year by a single person, or by three separate people on a single day. This is the conventional approach for working with this type of social media data because it corresponds to empirical user data collected by tourism sites that are often based on daily user access fees. For example, if three users access a park with daily user access fees on the same day, or a single user accesses that park on 3 separate days, both visitation scenarios appear identical in terms of empirical visitation rates (i.e. fees collected) as well as PUD.

## Pan-Arctic trends in the footprint of Arctic tourism over time

To quantify the 'footprint' of Arctic tourism (the percentage of the Arctic visited by tourists), we created a hexagonal spatial grid covering the entire Arctic with a 5km diameter resolution. We chose this resolution after examining the trade-off between commission errors and computing efficiency (S2 Appendix). We allocated each cell a value of 1 (visited) if any user had taken a photograph in that cell in a given year (2004–2017) and season (summer, winter), and 0 (unvisited) if not. The number of people using Flickr globally changed over time as the popularity of the platform waxed and waned. Relying on raw metrics of the number of Arctic Flickr users, photographs or photo-unit-days will thus result in biased estimates of tourism trends across time. The global and Arctic trends in Flickr use across time and the number of photos sampled in each year can be found in S3 Appendix. We calculated the footprint in three ways. Firstly, the *Uncorrected Arctic footprint* uses all available Flickr records (566205 summer; 228667 winter) and shows the full extent of Flickr users' footprint, but includes the bias from the global change in Flickr usage between 2004 and 2017. The *Global-bias corrected* subsample removes the global pattern of Flickr usage to represent a less source-biased view of the rise in the footprint of tourism in the Arctic. This was done by selecting a random sample of Arctic photos based on the change in number of photos submitted to Flickr globally using 2004 values as a baseline. For example, the year with the lowest global usage (2004), we kept all the photo records. If the global Flickr usage doubled in a particular year relative to 2004, we sampled half of the available records for that particular year. The number of records sampled for each month can be found in Table S3 Appendix (total 36546 summer records, 12262 winter). Finally, the *Equal sample size* also removes the global bias and provides a measure of the change in the relative footprint per-tourist across time. This was done by randomly selecting an equal number of photographs for each year from which to calculate the footprint (total 15652 records summer, 4018 records winter). Numbers for 2017 should be treated with caution and are likely underestimates as we extracted data from the Flickr API on 4 December 2017 and the average lag time between photos being taken and uploaded is 2 weeks, but can be longer [13].

## Modelling the influence of accessibility on seasonal patterns of Arctic landscape use

In order identify the effect of accessibility on the presence of tourists in different seasons, we first divided the study region into a hexagonal spatial grid with a 10km diameter resolution. We chose to use a lower resolution than that used in the footprint analysis above as the large number of cells at 5km made the models computationally intractable. We then calculated footprint in each cell for summer and for winter by allocating these cells a value of 1 if any photograph had been taken in that cell in that season (at any time from 2004 and 2017), and 0 if not, similar to the *Uncorrected Arctic footprint* described above. We removed cells that fell within Russia due to sparse coverage of this region by Flickr data, and from marine areas. We then modelled the footprint against variables describing the accessibility of each cell: log of distance to airports, log distance to ports, log distance to populated areas, log distance to road, the square root of the length of road within a grid cell, whether the cell overlapped any protected area, and country as a fixed effect. We chose these accessibility variables after examining a wider set of candidate variables for correlation. The models took the form of a binomial generalized additive model with logit link of footprint as a response variable. In order to account for spatial autocorrelation we included a fitted thin-plate spline on the variables latitude and longitude of each cell centroid. Smooths, intercept, slope and confidence intervals were estimated by a restricted maximum likelihood (REML) estimator and methods for large datasets ('bam'

function in R's mgcv package). The model formula is:

$$ln\left(\frac{P_{footprint}}{1 - P_{footprint}}\right)$$

$$= \beta_0 + f_1(long) + f_2(lat) + \beta_1 log_{10}(airport_{dist}) + \beta_2 log_{10}(port_{dist})$$

$$+ \beta_3 log_{10}(populated_{dist}) + \beta_4 log_{10}(road_{dist}) + \beta_5 \sqrt{road_{length}} + \beta_6 PA$$

$$+ \gamma_7 Country + \varepsilon$$

where $f_1, f_2$ are fitted thin-plate spline smooth functions on the variables latitude and longitude of each cell centroid, $airport_{dist}$ is the distance to airports, $port_{dist}$ is the distance to ports, $populated_{dist}$ is the distance to populated areas, $road_{dist}$ is the distance to road, $road_{length}$ is the length of road within a grid cell, and PA is whether the cell overlapped any protected area (0 if false, 1 if true). Random effects are represented by β's and the fixed effect is represented by .

The location of airports, ports and populated areas, and country boundaries were extracted from Natural Earth (www.naturalearthdata.com) using the R package 'rnaturalearth' [37]. Roads were extracted from Global Roads Inventory Project [38]. Protected area boundaries were drawn from CAFF [39] and supplemented with data from Protected Planet [40]. We ran separate models for summer and winter as the presence of snow and ice limits access to rural areas in winter. The summer model had 90,750 cells with no photos, and 6,554 with photos. The winter model had 93,990 cells with no photos, and 3,314 with photos.

### Local trends (booms and busts) in Arctic tourism

We investigated the suitability of Flickr data to be used to identify local booms and busts in tourism in the Arctic, a data deficient region. We first divided the landscape into a square grid cells and calculated the photo-unit-days in each cell for each year. We then performed linear regression modelling to identify trends in PUD in each cell between 2012 and 2017 and test their statistical significance. We ran models for cell diameters ranging from 500m to 100km to examine the effect of data availability on trend detection. We modelled only cells that were visited in at least two of the six years between 2012 and 2017. This timeframe was chosen as global Flickr usage remained reasonably constant during this period (Fig A in S3 Appendix).

Unless otherwise stated, all analysis was conducted in R version 3.4.2 [41] using the 'tidy-verse' [42], 'sf' [43], and 'mgcv' [44] packages. All spatial data was projected to North Pole Azimuthal Lambert equal area (EPSG:102017) for analysis. Code associated with the project is available at doi:10.18710/QEOFPY.

## Results

### Increase in Arctic summer and winter tourism footprint

We find that the overall footprint of tourism and area used per tourist has increased since 2004 (Fig 1). In the 10 years between 2006 and 2016, the uncorrected footprint increased from 0.066% to 0.385% of the Arctic in summer and 0.015% to 0.173% in winter. After correcting these figures to account for the increased proportion of tourists captured in the analysis over that time (i.e. accounting for the rise in Flickr use globally), the footprint increased by 374% in summer (0.029% of Arctic in 2006, 0.109% in 2016) and 634% in winter (0.006% of Arctic in 2006, 0.036% in 2016). The relative-footprint-per-tourist also increased over that period (summer: 0.028% of Arctic in 2006 to 0.043% in 2016; winter: 0.007% in 2006 to 0.012% in 2016). We caution that this metric is not the absolute per-tourist footprint, rather it should be interpreted as a relative indication of how the footprint of a fixed (and unquantified) number of

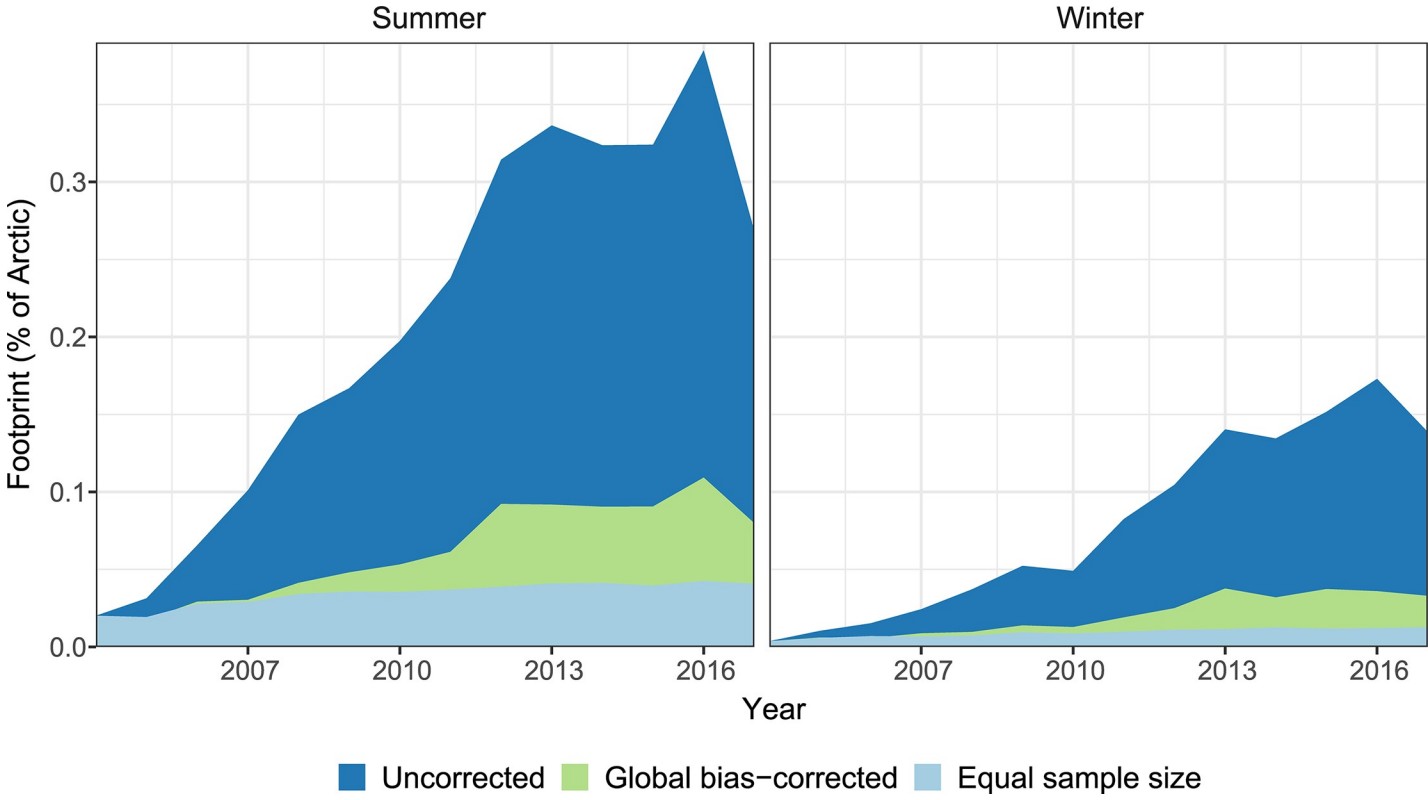

**Fig 1. Footprint of Arctic tourism.** The total footprint of Arctic tourism measured from Flickr data increased between 2004 and 2017 (*Uncorrected Arctic footprint*, darker blue), even after adjusting for the global rise in Flickr use during this period (*Global-bias corrected*, green). The relative footprint per tourist (*Equal sample size*, pale blue) also increased slightly over this time. Similar trends are seen in summer and winter, though the tourism footprint in winter is approximately half the magnitude of that in summer. The footprint is defined as the percentage of 5 km hexagonal grid cells within the Arctic region visited by at least one Flickr user per year. The 2017 decline should be interpreted with caution as it may in part be an artefact of the timing of our data download and the lag between photos being taken and their being uploaded to Flickr.

tourists has changed across time. These estimates are robust to uncertainty introduced by random sampling in the *Global-bias corrected* and *Equal sample size methods* (S4 Appendix).

### Growth in Arctic visitation over time

The total number of photos on Flickr increased between 2004 and 2017, both globally and in the Arctic. Global uploads of photos to Flickr steadily increased between 2004 and 2008, slowed between 2008 and 2012 before a slight upsurge in 2012, plateaued between 2013 and 2015 and has declined slightly since then (Fig A in S3 Appendix). The Arctic represents an increasing share of Flickr's yearly photo traffic (Table in S3 Appendix). The number of photos uploaded in the Arctic shows an exponential growth between 2004 and 2013, and has remained roughly steady since then (Fig B in S3 Appendix), with this trend overlaid on a seasonal trend in visitation (Fig C in S3 Appendix). Across the Arctic, July and August were the most popular months to photograph the Arctic (Fig D in S3 Appendix). Nonetheless, a large number of users visited during the Arctic winter (Table 1; 29.8% of all photos; 40,272 photo-unit-days, 53.3% of summer PUD). At the extreme, visitation in Greenland is concentrated in the summer months (91% of photos). In contrast to the rest of the Arctic, visitation in northern Finland peaks in winter (61.3% of photos). Winter and Christmas are a key part of the branding of tourism to northern Finland, and places such as 'Santa Claus's

**Table 1. Seasonal variation in Arctic visitation, estimated from Flickr data (2004 to 2017).**

| | Overall number of photos (% of all photos) | Summer (% of all photos for that region) | Winter (% of all photos for that region) | Number of photo-unit-days (PUD) | Summer (% of total PUD for that region) | Winter (% of total PUD for that region) |
|---|---|---|---|---|---|---|
| Whole Arctic | 805,684 | 70.2 | 29.8 | 115,775 | 65.2 | 34.8 |
| Iceland | 377,817 (46.9) | 70.0 | 30.0 | 51,500 | 66.4 | 33.6 |
| Alaskan Arctic | 150,325 (18.7) | 76.0 | 24.0 | 21,903 | 67.0 | 33.0 |
| Norwegian Arctic | 122,387 (15.2) | 68.2 | 31.8 | 20,779 | 63.7 | 36.3 |
| Canadian Arctic | 43,466 (5.4) | 68.4 | 31.6 | 7,713 | 66.8 | 33.2 |
| Finnish Arctic | 22,164 (2.8) | 38.7 | 61.3 | 3,716 | 43.2 | 56.8 |
| Faroe Islands | 20,259 (2.5) | 78.1 | 21.9 | 2,226 | 68.0 | 32.0 |
| Greenland | 19,944 (2.5) | 91.0 | 9.0 | 1,952 | 78.8 | 21.2 |
| Swedish Arctic | 18,303 (2.3) | 53.5 | 46.5 | 2,809 | 56.9 | 43.1 |
| Svalbard & Jan Mayen | 16,282 (2.0) | 71.3 | 28.7 | 1,321 | 72.6 | 27.4 |
| Russian Arctic | 13,321 (1.7) | 64.2 | 35.8 | 2,228 | 60.7 | 39.3 |

village' in Rovaniemi, Lappland, attract the majority of the visitors to the region (Fig 2). These metrics of Arctic visitation derived from Flickr show good agreement with official metrics of tourism visitation (S1 Table), consistent with previous research on social media data [12,15,21].

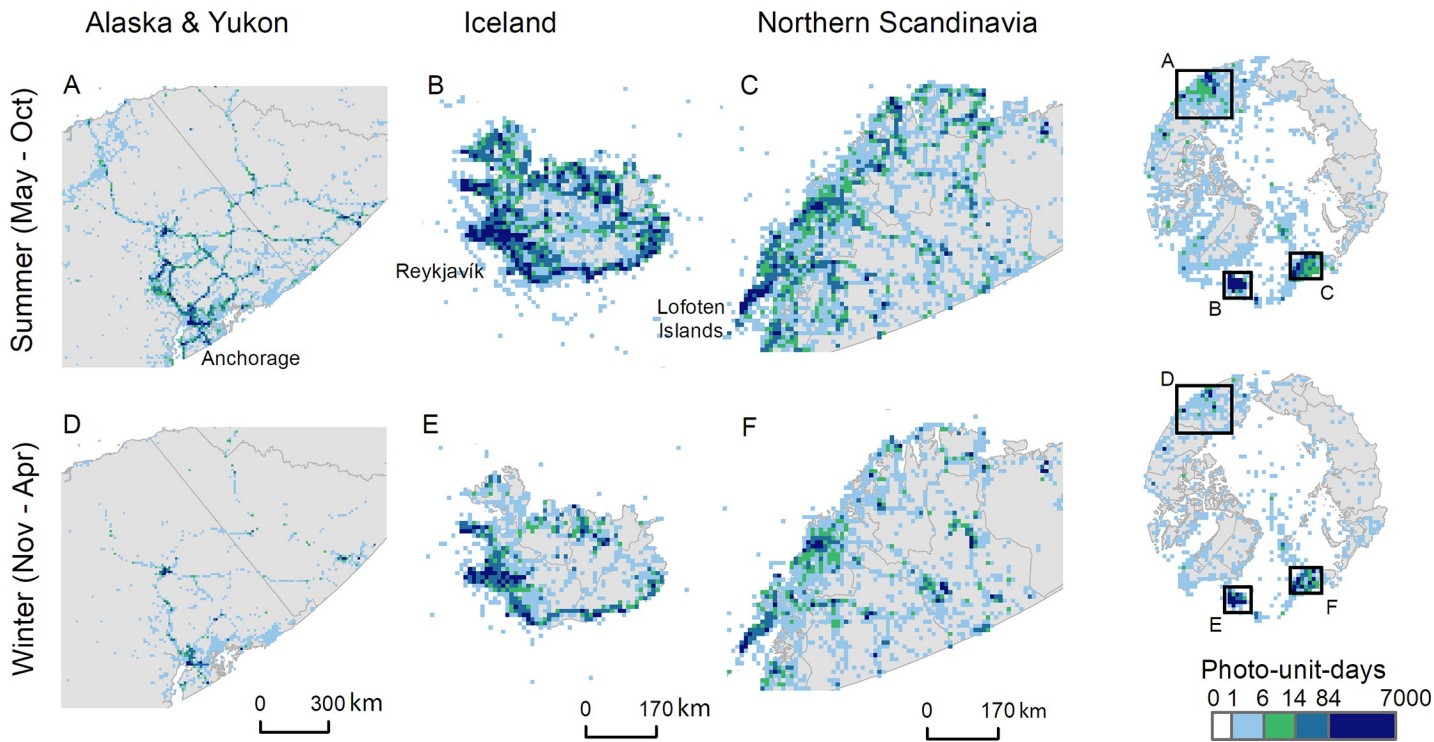

**Fig 2. Seasonal maps of Arctic tourism (2004–2017) displayed at 10km resolution.** The guide maps (right) are displayed at 100km resolution. Spatial patterns of tourism are strongly governed by air, road and sea access, with few tourists venturing far from populated areas in winter. A photo-unit-day value of 14 corresponds to one Flickr user visiting the cell per year. Country borders are modified from Natural Earth CC PD.

## Flickr data not suitable as early-warning system

We investigated the potential for Flickr data to be used to identify local trends in tourism over time. The Arctic is a data-poor region and we found that only a small number of places were photographed more than once per year (Fig in S2 Appendix). Only 22 10km grid cells were visited by Flickr users more than 50 times (i.e. once a week) in 2017. Although annual tourism growth is documented as ranging from 5–20% across the Arctic, due to these data limitations regression models of trends in visitation between 2012 and 2017 were able to detect significant trends at just a handful of sites (S1 Fig). Most sites lacked sufficient data to detect trends even when aggregated to 10km grid cells, a relatively large scale compared to the scale required for management decisions.

## Spatial patterns of Arctic visitation change across the seasons

The spatial pattern of tourists' landscape use differed between summer and winter. Flickr users ventured further north and into marine areas to a greater extent during the summer months (Fig 2). Visitation was influenced by access and often, though not always, concentrated in recognized tourism hotspots (Fig 2). The main hotspots of tourism fall along coastal roads in Iceland, in the fjords and islands of northern Norway, and in protected areas and along roads in North America (Fig 2). We note that although the size of the Alaskan tourism market eclipses that of the rest of the Arctic, including Iceland, few cruises travel further north than Anchorage (61.2˚N), and the majority of this tourism thus falls outside our study region which is bounded to the south at 60˚N.

## Accessibility drives Arctic visitation

We found that accessibility has a significant effect on the distribution of Flickr users throughout the Arctic, with the presence of tourists decreasing rapidly as distance from transport infrastructure and populated areas increases, and increasing with the length of road in a given cell (Table 2). The summer accessibility model explained 47.3% of the deviance in the presence of tourists (adjusted $R^2$ 0.448, AIC 25347). The winter accessibility model explained 51.4% of the deviance in the presence of tourists (adjusted $R^2$ = 0.436, AIC of 14117). The variable *square root of length of road* in cell had the largest explanatory power of the accessibility variables (model without this variable had adjusted $R^2$ 0.409 summer, 0.393 winter, deviance explained 44.7% summer, 48.8% winter). The protected area term of the winter model was significant (p = 0.000670) in the full model, but not significant in more parsimonious models. Removal of this term only slightly decreased the deviance explained ($\Delta$AIC = 9, $\Delta$df = -1.0239, $\Delta$deviance = -10.706, Pr(>Chi) = 0.001117), indicating that tourists were no more or less likely to visit protected areas than non-protected areas in winter. All other terms were significant in both models. We removed the variable that had the least explanatory power, *log distance to port*, from the summer model, and the winter model without the protected area term. This reduced the model fit of the summer model ($\Delta$AIC = 145, $\Delta$df = -1.0881, $\Delta$deviance = -146.57, Pr(>Chi) = 2.066x10$^{-16}$) but had minimal effect on the winter model ($\Delta$AIC = 25, $\Delta$df = -1.1028, $\Delta$deviance = -27.406), though the large number of data points in the model meant that this variable was significant at 95% confidence (2.02x10$^{-7}$). Removal of all other terms substantially decreased the deviance explained by the models. Visual examination of model residuals did not reveal any residual spatial autocorrelation in any of the models. Plots of model fit and partial plots of model variables are included in S5 Appendix.

**Table 2. Model coefficients for binomial generalized additive models of the effect of accessibility on the tourist footprint in summer and winter in the Arctic.** The intercept represents Norway, unprotected. The protected area term was not included in the winter model.

| Variable | Summer | | | Winter | | |
|---|---|---|---|---|---|---|
| | Estimate of coefficient (±standard error) | z-value | Pr(>\|z\|) | Estimate of coefficient (±standard error) | z-value | Pr(>\|z\|) |
| (Intercept) | 8.720 ±0.619 | 14.094 | $<2.00 \times 10^{-16}$ | 7.830 ±0.925 | 8.468 | $<2.00 \times 10^{-16}$ |
| Norway | | | | | | |
| Canada | -5.707 ±0.632 | -9.035 | $<2.00 \times 10^{-16}$ | -5.690 ±0.961 | -5.923 | $3.17 \times 10^{-9}$ |
| Finland | -0.474 ±0.159 | -2.988 | 0.0028 | 0.979 ±0.171 | 5.728 | $1.01 \times 10^{-8}$ |
| Faroe Islands | 5.802 ±22.647 | 0.256 | 0.798 | -5.557 ±0.970 | -5.731 | $1.00 \times 10^{-8}$ |
| Greenland | -5.167 ±0.590 | -8.757 | $<2.00 \times 10^{-16}$ | -4.805 ±0.893 | -5.381 | $7.42 \times 10^{-8}$ |
| Iceland | -4.082 ±0.624 | -6.540 | $6.15 \times 10^{-11}$ | -3.712 ±0.914 | -4.063 | $4.85 \times 10^{-5}$ |
| Svalbard & Jan Mayen | -0.059 ±0.267 | -0.221 | 0.825 | -1.524 ±0.390 | -3.939 | $8.18 \times 10^{-5}$ |
| Sweden | -0.673 ±0.134 | -5.007 | $5.52 \times 10^{-07}$ | 0.022 ±0.150 | 0.147 | 0.883 |
| USA (Alaska) | -4.682 ±0.630 | -7.433 | $1.06 \times 10^{-13}$ | -5.647 ±0.954 | -5.917 | $3.28 \times 10^{-9}$ |
| Protected area TRUE | 0.841 ±0.042 | 19.919 | $<2.00 \times 10^{-16}$ | | | |
| Square root of length of road in cell | 0.805 ±0.026 | 31.326 | $<2.00 \times 10^{-16}$ | 0.574 ±0.023 | 24.717 | $<2.00 \times 10^{-16}$ |
| Log distance to road | -0.278 ±0.019 | -14.912 | $<2.00 \times 10^{-16}$ | -0.278 ±0.025 | -11.217 | $<2.00 \times 10^{-16}$ |
| Log distance to airports | -0.220 ±0.027 | -8.135 | $4.13 \times 10^{-16}$ | -0.337 ±0.034 | -9.855 | $<2.00 \times 10^{-16}$ |
| Log distance to ports | -0.381 ±0.031 | -12.481 | $<2.00 \times 10^{-16}$ | -0.229 ±0.038 | -5.970 | $2.37 \times 10^{-9}$ |
| Log distance to populated places | -0.532 ±0.022 | -24.243 | $<2.00 \times 10^{-16}$ | -0.603 ±0.028 | -21.244 | $<2.00 \times 10^{-16}$ |

## Discussion

The footprint of tourism on the Arctic environment has almost quadrupled over the past decade, from 0.03% of the Arctic in summer 2006 to 0.11% of the Arctic in summer 2016 (Fig 1), and the winter footprint has increased by over 600%. Despite this dramatic expansion, large areas of the Arctic still remain free from tourists (Fig 2). Arctic tourists tend to congregate in a handful of highly visited sites (Fig 2) with a long tail of places that are visited only occasionally, following the power law seen in other parts of the world [45]. The recent tourism boom across the Arctic has led to widespread concerns over the effects of tourism on Arctic ecosystems and communities, and the sustainability of Arctic tourism [4,5]. Arctic tourism is often marketed around ideas of pristine, untouched nature, and the overall growth in tourist numbers and the concurrent increase in infrastructure to support that growth presents challenges for maintaining environmental and social sustainability. There are indications that many popular sites are reaching capacity, and that tourists are beginning to experience disappointment and frustration around the large number of visitors present [46]. The footprint per tourist has also increased (Fig 1), indicating that tourists are now visiting a wider variety of places. This may be either due to self-segregation by tourists seeking an undisturbed experience [46], or the marketing of a wider variety of tourist attractions and nature-based activities as Arctic tourism matured over the past decade [47].

These patterns of visitation present both advantages for the management of tourists and challenges for the sharing of the economic benefits of the recent and ongoing Arctic tourism

boom. Management of tourists and their impacts is often easier where they congregate in a few small areas as resources can be allocated to these high priority areas bringing economies of scale. This is particularly important in small, rural communities where both human and financial capacity to manage tourist impacts can be limited [24]. Though environmental impacts can be locally high in these well-visited areas, requiring thoughtful management, aggregation of tourists in a few small areas can reduce the impacts of tourism across the wider landscape and channel resources into efficient management at the high use sites to sustain tourist satisfaction despite crowding. Channeling visitors into such 'sacrificial sites' may have net benefits at the landscape scale in places like the Arctic, where wildlife and vegetation are highly sensitive to disturbance and take a long time to recover from human impacts.

The tourism footprint is strongly associated with access, and is particularly dependent on the presence of roads and airports. This is not surprising, but important to keep in mind when planning for growth in tourism. The proposed construction of three transatlantic airports in Greenland has a high potential of boosting tourism in host communities that do not have sufficient capacity to sustainably manage this growth. The sustainability of tourism in Greenland, and similar developing places around the world depends heavily on building community capacity that sustains natural resources and local culture, and that protects vulnerable sites and species from the expansion of tourists into new locations [6]. The spatial extent of the winter footprint of tourism is about 40% lower than that in summer, with tourists gathering closer to airports and towns. Managers therefore need to be especially aware of the expansion of tourists into vulnerable sites and the greater use of protected areas in the summer season. Ports have only minimal influence on the presence of tourists on land.

While social media data may be useful for rapidly detecting localized booms in tourism in highly visited regions [48], our analysis indicates that Flickr data is of limited use for identifying local tourism booms in data deficient regions such as in the Arctic. Low rates of visitation across most of the Arctic combined with the small proportion of visitors that use Flickr [15] means that just a handful of Arctic locations are visited by Flickr users more than once a month (Fig C in S1 Fig). One of the few regions where we detected statistically significant increases in visitation was the Lofoten islands of northern Norway. There, our analysis confirmed qualitative trends already noticed by tourism agencies in the region. Twitter has a higher user base and may be a better source of fine-scale temporal data in the parts of the Arctic with high-speed mobile data coverage [18,49,50]. Changes in ownership of the social media platforms and changes to data access rules means social media data from other suitable platforms such as Panoramio and Instagram were no longer freely available to academic researchers at the time of analysis. Quantitative analysis of social media data requires specialized computing and technical skills that are not normally available to local tourism management agencies. Maintaining dialogue between tourism management bodies and local communities and tour operators remains the most pragmatic way to detect and respond to fine-scale tourism trends in areas where data and technical capacity are limited.

## Conclusions

The recent and rapid increase in the footprint of tourists on the Arctic that we document here is concerning. Upcoming investments in transport infrastructure in places like Greenland and the promotion of remote areas of the Arctic as tourist destinations, such as Franz Josef in Russia, will drive a further expansion of the tourist footprint in this unique part of the world. Destinations are also increasingly been marketed as 'last chance tourism' attracting visitors to venture into previously unexplored areas to experience Arctic ecosystems and species at risk of disappearing [4,51]. For instance, in Hudson Bay in Canada the small community of Churchill

where polar bears spend increasingly more time on shore due to climate change, have experienced a rapid influx of tourists [4]. Wildlife viewing of vulnerable species, such as polar bears, narwhals and beluga whales, is putting additional pressure on species threatened by climate change [52,53]. Our high resolution and seasonal maps of Arctic tourism allow tourist management bodies and environmental organizations to pinpoint the places visited by tourists and the relative magnitude of visitation across the Arctic and to detect landscape-wide trends in visitation that need to be managed. Strategic and thoughtful assessment of whether this ongoing growth in Arctic tourism is sustainable or desirable for Arctic ecosystems and communities is urgently required.

## Supporting information

**S1 Appendix. Sensitivity analysis of photo exclusion threshold.**
(PDF)

**S2 Appendix. Sensitivity analysis of resolution.**
(PDF)

**S3 Appendix. Global and Arctic Flickr trends.**
(PDF)

**S4 Appendix. Uncertainty around footprint estimates.**
(PDF)

**S5 Appendix. GAM model performance.**
(PDF)

**S1 Fig. Annual maps of tourism growth.**
(PDF)

**S1 Table. Comparison with official visitor statistics.**
(PDF)

## Author Contributions

**Conceptualization:** Claire A. Runge, Vera H. Hausner.

**Data curation:** Claire A. Runge.

**Formal analysis:** Claire A. Runge, Remi M. Daigle.

**Funding acquisition:** Vera H. Hausner.

**Methodology:** Claire A. Runge, Remi M. Daigle.

**Software:** Remi M. Daigle.

**Visualization:** Claire A. Runge, Remi M. Daigle.

**Writing – original draft:** Claire A. Runge.

**Writing – review & editing:** Claire A. Runge, Remi M. Daigle, Vera H. Hausner.

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
