## [Decision Letter · Decision Letter 0]

18 Aug 2019

PONE-D-19-16929

Quantifying tourism booms and the increasing footprint in the Arctic with social media data

PLOS ONE

Dear Dr Runge,

Thank you for submitting your manuscript to PLOS ONE. After careful consideration, we feel that it has merit but does not fully meet PLOS ONE’s publication criteria as it currently stands. Therefore, we invite you to submit a revised version of the manuscript that addresses the points raised during the review process.

Both reviewers see the merits of the study reported in the manuscript. I would encourage you to address their comments in detail. 

We would appreciate receiving your revised manuscript by Sep 27 2019 11:59PM. To enhance the reproducibility of your results, we recommend that if applicable you deposit your laboratory protocols in protocols.io, where a protocol can be assigned its own identifier (DOI) such that it can be cited independently in the future. For instructions see: http://journals.plos.org/plosone/s/submission-guidelines#loc-laboratory-protocols

We look forward to receiving your revised manuscript.

Kind regards,

Wenwu Tang

Academic Editor

PLOS ONE

Journal Requirements:

Please add a statement in the revised manuscript confirming you complied by the Terms & conditions of Flickr when gathering the data for the study.

Reviewers' comments:

Reviewer's Responses to Questions

**Comments to the Author**

1. Is the manuscript technically sound, and do the data support the conclusions?

Reviewer #1: Partly

Reviewer #2: Yes

2. Has the statistical analysis been performed appropriately and rigorously? 

Reviewer #1: Yes

Reviewer #2: Yes

3. Have the authors made all data underlying the findings in their manuscript fully available?

Reviewer #1: Yes

Reviewer #2: Yes

4. Is the manuscript presented in an intelligible fashion and written in standard English?

Reviewer #1: Yes

Reviewer #2: Yes

5. Review Comments to the Author

Reviewer #1: The current study attempts to explore the spatiotemporal patterns of visitations in the Arctic with social media data. This work is an interesting topic, which is supposed to benefit the tourist management and ecosystem conservation planning. 

However, due to the poorly-organized methodology and some other concerns, I recommended major revision of this manuscript. The comments are listed in the following. 

(1) My major concern is about the scaling effects on the results in this study. In the context of sparse data, how does the spatial resolution affect the results, or the trend of the footprint? did the authors attempt to examine the scaling effect? 

(2) Line 119-142, the sample size is the key to representative samples, so what is the specific sample size in three ways to calculate the footprint? The authors selected an random sampling method over time, what about the spatial pattern of the sampled photographs? The authors may examine the effects of the spatial pattern of the sampled records on the analysis outcomes.

(3) Line 155, how did the authors examine the correlations of candidate variables and seasonal patterns of the footprint? Does the multicollinearity exist? 

(4) Line 156-160, data collection and processing should be separated from the methodology part. 

(5) Line 161-163, a mathematical model is highly recommended in this part to describe the relationship between the presence of tourists and the accessibility factors, I think it will make the model description more clear. Actually, the methodology section should emphasize the model formulation, not only the analysis procedures. Besides, an analysis flowchart will make this part flow more logically. 

(6) The language of this study should be further improved. Some spelling and grammar mistakes should be carefully corrected, for example, Line 144:”in order identify”, Line 270: the format of Table 2 is not consistent with Table 1, Line 328: the format of “Conclusion”.

Reviewer #2: This paper leverages social media data(geotagged flickr photos) to quantify the tourism booms and visitation patterns in the Arctic area. Their results indicate that social media cannot be used as an early warning system of tourism growth in the study area but is able to capture the growth and seasonal patterns of the visitations. Spatial statistics and modelling is used in the analysis.

I think this is a high quality research and very well-written paper The structure is clear, logic is smooth, and language is easy to understand and almost free of typos or syntax errors. Enough details are given for both data collection and processing as well as the steps of the analysis, which make it possible to reproduce the findings. I also appreciate the authors cautions in interpreting some of the results and warn readers about the potential bias etc. This paper is close to be ready for publication after addressing some minor issues as below:

1. Line 102 to 105, why only exclude those with only 1 or 2 photos? People with 3 photos are considered as tourist not test users? More explanation about the rational of setting these numbers is needed here. Could you show the distribution diagram of the number of users having number of photos? (I guess it should show a long-tail diagram?)

2. I am not convinced that PUD is a good measure here. As authors stated "three PUD can represent either a site visited on three separate days throughout the year by a single person, or by three separate people on a single day.".I don't think "a site visited on three separate days throughout the year by a single person" and "a site visited by three separate people on a single day" indicates the same visitation activity. I would like to see some justification of this in the revision.

3. For spatial aggregation, the authors used both "hexagonal spatial grid" and "square grid", why not only use one type to make it consistent?

4. Line 144: "In order" change to "In order to".

6. PLOS authors have the option to publish the peer review history of their article (what does this mean?). If published, this will include your full peer review and any attached files.

Reviewer #1: No

Reviewer #2: No

---

## [Author Response · Author response to Decision Letter 0]

23 Oct 2019

Response to Review Comments to the Author

Reviewer #1: The current study attempts to explore the spatiotemporal patterns of visitations in the Arctic with social media data. This work is an interesting topic, which is supposed to benefit the tourist management and ecosystem conservation planning. 

However, due to the poorly-organized methodology and some other concerns, I recommended major revision of this manuscript. The comments are listed in the following. 

Response: We include detailed responses to each of the reviewer’s comments below.

(1) My major concern is about the scaling effects on the results in this study. In the context of sparse data, how does the spatial resolution affect the results, or the trend of the footprint? did the authors attempt to examine the scaling effect? 

Response: Yes, we examined the scaling effect and this is detailed in S2 Appendix. Our sensitivity analysis indicated that the footprint remained reasonably constant up to approximately 25 km2 cell area. Thus for the footprint analysis, 5 km diameter hexagons (21.6 km2 cell area) were an appropriate compromise between overestimating the footprint (commission errors) while keeping the number of cells low enough that computations could be undertaken with the memory available to us. For temporal trend estimates, sensitivity analysis indicated that 10 km diameter hexagonal cells (86.6 km2 cell area) seems a good compromise between providing spatial resolution at a scale relevant for management and having sufficient data in each cell to analyse trends.

(2) Line 119-142, the sample size is the key to representative samples, so what is the specific sample size in three ways to calculate the footprint? The authors selected an random sampling method over time, what about the spatial pattern of the sampled photographs? The authors may examine the effects of the spatial pattern of the sampled records on the analysis outcomes.

Response: We now include the total sample size for each method in the manuscript which we previously included by month and season in Table S2 (now Table S3). The reviewer is correct, random subsamples used in two of the three methods to calculate the footprint will generate differences in the spatial patterning between random draws. We performed a sensitivity analysis of the uncertainty introduced by random sampling which we now include as Appendix S4. The results indicate that our findings are robust to the uncertainty introduced by sampling. 

(3) Line 155, how did the authors examine the correlations of candidate variables and seasonal patterns of the footprint? Does the multicollinearity exist? 

Response: As we state on L162 we retained the specified set of variables which showed low multicollinearity, after examining a wider set of candidate variables for collinearity.

(4) Line 156-160, data collection and processing should be separated from the methodology part. 

Response: We respectfully disagree, and believe that summary of the methods used in data collection and processing is best described within the methodology section. 

(5) Line 161-163, a mathematical model is highly recommended in this part to describe the relationship between the presence of tourists and the accessibility factors, I think it will make the model description more clear. Actually, the methodology section should emphasize the model formulation, not only the analysis procedures. Besides, an analysis flowchart will make this part flow more logically. 

Response: The model follows the general formulation below, which we now include in the manuscript on L169:

g(footprint) =β_0+ f_1 (long) + f_1 (lat) +β_1 〖log〗_10 (airport_dist) + β_2 〖log〗_10 (port_dist) +β_2 〖log〗_10 (popula〖ted〗_dist) + β_3 〖log〗_10 (〖road〗_dist) 

+ β_4 √(〖road〗_length ) +β_5 PA +γ_6 Country +ε 

where g is the logit link function, f1,f2 are smooth functions estimated by the model by REML, airportdist is the distance to airports, portdist is the distance to ports, populateddist is the distance to populated areas, roaddist is the distance to road, roadlength is the length of road within a grid cell, and PA is whether the cell overlapped any protected area (0 if false, 1 if true). Random effects are represented by β’s and the fixed effect is represented by ℽ.

(6) The language of this study should be further improved. Some spelling and grammar mistakes should be carefully corrected, for example, Line 144:”in order identify”, Line 270: the format of Table 2 is not consistent with Table 1, Line 328: the format of “Conclusion”.

Response: Done. We note that the journal has a typesetting process for tables post review. Conclusions is a subheading within Discussion as suggested in the journal guidelines, hence the different formatting. 

Reviewer #2: This paper leverages social media data(geotagged flickr photos) to quantify the tourism booms and visitation patterns in the Arctic area. Their results indicate that social media cannot be used as an early warning system of tourism growth in the study area but is able to capture the growth and seasonal patterns of the visitations. Spatial statistics and modelling is used in the analysis.

I think this is a high quality research and very well-written paper The structure is clear, logic is smooth, and language is easy to understand and almost free of typos or syntax errors. Enough details are given for both data collection and processing as well as the steps of the analysis, which make it possible to reproduce the findings. I also appreciate the authors cautions in interpreting some of the results and warn readers about the potential bias etc. This paper is close to be ready for publication after addressing some minor issues as below:

Response: We thank the reviewer for their positive reception of our manuscript.

1. Line 102 to 105, why only exclude those with only 1 or 2 photos? People with 3 photos are considered as tourist not test users? More explanation about the rational of setting these numbers is needed here. Could you show the distribution diagram of the number of users having number of photos? (I guess it should show a long-tail diagram?)

Response: We examined the characteristics of different types of users during the data cleaning process we undertook before analysing this dataset. When we examined the photographs of a sample of ‘test users’ (those with 1 or 2 photos) these appeared to be ‘random snaps’ i.e. out of focus or with no discernable subject. Thus we felt justified in excluding these users. The reviewer makes a good point about why choose 2 photos as a cutoff point, and we acknowledge this threshold is somewhat arbitrary one that attempts to balance retaining quality photos without excluding too many users. People contributing 10 or fewer photos account for less than 1% of the number of photographs, but over two thirds of the users. This dropped to one third of users with 2 or fewer photos. We now include the histogram and discussion of this issue in Appendix S1. The median number of photos per user ranges across countries from 25-33 photos. 

2. I am not convinced that PUD is a good measure here. As authors stated "three PUD can represent either a site visited on three separate days throughout the year by a single person, or by three separate people on a single day.".I don't think "a site visited on three separate days throughout the year by a single person" and "a site visited by three separate people on a single day" indicates the same visitation activity. I would like to see some justification of this in the revision.

Response: This is the conventional approach for working with this type of social media data because it corresponds to empirical user data collected by tourism sites that are often based on daily user access fees. For example, if three users access a park with daily user access fees on the same day, or a single user accesses that park on 3 separate days, in terms of empirical visitation rates (i.e. fees collected) as well as PUD both visitation scenarios appear identical. We now clarify this in the text on L119.

3. For spatial aggregation, the authors used both "hexagonal spatial grid" and "square grid", why not only use one type to make it consistent?

Response: The two grid types each have benefits for the two different analyses. Hexagonal grid cells minimise spatial biases introduced by gridding data, and we regarded as a better choice for the footprint analyses than square grid cells. Ideally we would have used hexagonal grid cells for both analyses, however the large number of cells meant that for the model analysis we needed to take advantage of the computational efficiencies available from raster processing tools. 

4. Line 144: "In order" change to "In order to".

Response: Done

---

## [Decision Letter · Decision Letter 1]

16 Dec 2019

Quantifying tourism booms and the increasing footprint in the Arctic with social media data

PONE-D-19-16929R1

Dear Dr. Runge,

We are pleased to inform you that your manuscript has been judged scientifically suitable for publication and will be formally accepted for publication once it complies with all outstanding technical requirements.

With kind regards,

Wenwu Tang

Academic Editor

PLOS ONE

Additional Editor Comments (optional):

Reviewers' comments:

Reviewer's Responses to Questions

**Comments to the Author**

1. If the authors have adequately addressed your comments raised in a previous round of review and you feel that this manuscript is now acceptable for publication, you may indicate that here to bypass the “Comments to the Author” section, enter your conflict of interest statement in the “Confidential to Editor” section, and submit your "Accept" recommendation.

Reviewer #1: All comments have been addressed

Reviewer #2: All comments have been addressed

2. Is the manuscript technically sound, and do the data support the conclusions?

Reviewer #1: Yes

Reviewer #2: Yes

3. Has the statistical analysis been performed appropriately and rigorously? 

Reviewer #1: Yes

Reviewer #2: Yes

4. Have the authors made all data underlying the findings in their manuscript fully available?

Reviewer #1: Yes

Reviewer #2: Yes

5. Is the manuscript presented in an intelligible fashion and written in standard English?

Reviewer #1: Yes

Reviewer #2: Yes

6. Review Comments to the Author

Reviewer #1: (No Response)

Reviewer #2: (No Response)

7. PLOS authors have the option to publish the peer review history of their article (what does this mean?). If published, this will include your full peer review and any attached files.

Reviewer #1: No

Reviewer #2: No

---

## [Editor Report · Acceptance letter]

20 Dec 2019

PONE-D-19-16929R1 

Quantifying tourism booms and the increasing footprint in the Arctic with social media data 

Dear Dr. Runge:

I am pleased to inform you that your manuscript has been deemed suitable for publication in PLOS ONE. Congratulations! Your manuscript is now with our production department. 

With kind regards,

on behalf of

Dr. Wenwu Tang 

Academic Editor

PLOS ONE